# Higher-Quality Pumpkin Cultivars Need to Recruit More Abundant Soil Microbes in Rhizospheres

**DOI:** 10.3390/microorganisms10112219

**Published:** 2022-11-09

**Authors:** Yan Sun, Ziyue Huang, Siyu Chen, Da Yang, Xinru Lin, Wenjun Liu, Shangdong Yang

**Affiliations:** 1Guangxi Key Laboratory of Agro-Environment and Agro-Products Safety, National Demonstration Center for Experimental Plant Science Education, Agricultural College, Guangxi University, Nanning 530004, China; 2Vegetable Research Institute, Guangxi Academy of Agricultural Sciences, Nanning 530007, China

**Keywords:** pumpkin (*Cucurbita moschata* Duchesne), quality, rhizosphere, microbial compositions

## Abstract

Two different qualities of pumpkin, cultivars G1519 and G1511, were grown in the same environment under identical management. However, their qualities, such as the contents of total soluble solids, starch, protein, and vitamin C, were significantly different. Do rhizospheric microbes contribute to pumpkin quality? To answer this question, this study investigated the soil microbial compositions in the rhizospheres of different quality pumpkin cultivars to determine the differences in these soil microbial compositions and thus determine how soil microbes may affect pumpkin quality. Firstly, a randomized complete block design with two pumpkin cultivars and three replications was performed in this study. The soil microbial compositions and structures in the rhizospheres of the two pumpkin cultivars were analyzed using a high-throughput sequencing technique. In comparison with the low-quality pumpkin cultivar (G1519), higher microbial diversity and richness could be found in the rhizospheres of the high-quality pumpkin cultivar (G1511). The results showed that there were significant differences in the soil bacterial and fungal community compositions in the rhizospheres of the high- and low-quality pumpkin cultivars. Although the compositions and proportions of microorganisms were similar in the rhizospheres of the two pumpkin cultivars, the proportions of Basidiomycota and *Micropsalliota* in the G1519 rhizosphere were much higher than those in the G1511 rhizosphere. Furthermore, the fungal phylum and genus Rozellomycota and *Unclassified_p__Rozellomycota* were unique in the rhizosphere of the high-quality pumpkin cultivar (G1511). All the above results indicate that soil microbes were enriched differentially in the rhizospheres of the low- and high-quality pumpkin cultivars. In other words, more abundant soil microbes were recruited in the rhizosphere of the high-quality pumpkin cultivar as compared to that of the low-quality cultivar. Rozellomycota and *Unclassified_p__Rozellomycota* may be functional microorganisms relating to pumpkin quality.

## 1. Introduction

Pumpkin (*Cucurbita moschata* Duch.) is an annual vegetable crop widely distributed across most climate regions [1]. Some *Cucurbita* melon vegetables, particularly pumpkins, are important nutrient sources in less-developed countries in tropical areas [1]. Pumpkin is a nutritious vegetable that meets the requirements of healthy nutrition [2]. Pumpkin fruits are rich in soluble solids, starch, vitamin C, proteins, and additional biologically active substances and nutrients [3]. The further increase in the quality of pumpkins is a common goal pursued by consumers, processors, and breeders [4].

The rhizosphere is described as the most active interface on Earth since it is home to numerous living microorganisms and invertebrates [5]. Soil microbes are involved in many critical ecosystem processes, such as nutrient acquisition, biogeochemical cycling [6], and soil aggregation [7], and they play a significant role in the development of sustainable agriculture [8,9,10]. In addition, soil microbes strongly influence plant productivity through direct or indirect effects [8]. Soil microorganisms are affected by environmental conditions, soil characteristics, plant species, and crop management regimes [11,12,13,14].

Numerous studies have been conducted on the diversity of the rhizosphere’s microbial community, with a special focus on cultivated plants [15,16,17]. Cultivated varieties may differ in terms of root exudation [18], since crops influence soil microbial diversity and abundance through the type and amount of root exudate released from them [19,20,21,22,23]. Somers et al. [24] reported that the abundance of rhizosphere microorganisms is affected by the composition of root exudates and the compositions of root exudates were also derived from crop cultivars. Therefore, it is obvious that different crop varieties have different soil microbes in root colonization [24,25,26] and the rhizosphere microbial communities of some cultivars might be unique [24,26,27,28].

Microbial compositions and their functions in rhizospheres have been thoroughly studied for more than a century to ascertain the effects of plant species, cultivars, and soil types on them [29,30,31,32,33]. Experimental studies showed cultivar-specific selection in rhizosphere communities [22,33,34]. Recently, most of the studies on crop microorganisms have focused on the associations between cultivar morphologies and the microbial community structures in rhizospheres, particularly their antagonistic potential versus diseases [35,36,37]. Yao and Wu [38] reported that higher bacterial diversity and abundance in two fusarium-wilt-resistant cultivars of cucumbers could be detected, as compared to susceptible cultivars. Han et al. [39] indicated that three fox-tail millet cultivars with different smut resistance resided in various rhizosphere bacterial communities and the suppression of smut disease was associated with high bacterial diversity in the rhizosphere. Wei et al. [40] demonstrated that specific groups of rhizosphere microbiota and root endophytes might be related to cotton’s resistance to Verticillium dahliae. Most studies on the effects of crop cultivars on rhizosphere microorganisms have only focused on the differences in rhizosphere microorganism varieties in association with different resistance levels to soil-borne diseases. However, there is a lack of information to signify whether plant cultivars with different nutrients select different microbial communities in the rhizosphere.

We used high-throughput sequencing technology to understand the potential associations of rhizosphere soil microbial communities with fruit nutrients. The purpose of this study was to examine whether soil microbial community structures in the rhizospheres of pumpkin cultivars differ with high and low nutrient levels. If differences are found concerning the cultivar, this might be used to exploit plant germplasms stemming from crop diversification. Moreover, this knowledge would advance our comprehension of the connection between the soil microbiome and crop nutrition. We hypothesized that there would be significant differences among pumpkin cultivars with different fruit nutrients in their microbial communities. This result would indicate that pumpkins with high nutrient levels would have a stronger response, including higher diversity and abundance of soil microbes. Furthermore, crop cultivars would be one of the significant factors determining the rhizobacterial community composition.

## 2. Materials and Methods

### 2.1. Experimental Conditions and Plant Materials

The experiment was performed at the Lijian Scientific Research Base of GXAAS in Nanning, Guangxi Zhuang Autonomous Region, China (23°14′ N 108°02′ W).

The soils at the test site were all acid red loams with total nitrogen content of 0.81 g kg^−1^, total phosphorus content of 0.39 g kg^−1^, total potassium content of 2.68 g kg^−1^, available nitrogen content of 53.7 mg kg^−1^, available phosphorus content of 9.1 mg kg^−1^, available potassium content of 89.0 mg kg^−1^, organic matter content of 12.9 g kg^−1^, and pH of 5.46.

This study was conducted using two pumpkin cultivars with comparatively large planting areas in Guangxi Zhuang Autonomous Region (GZAR), China, known as G1511 and G1519. The pumpkins were planted at Lijian Scientific Research Base of GXAAS on 9 September and they were harvested on 19 December 2019. The experimental fields were identically managed during planting. The quality characteristics of the two cultivars of pumpkin are shown in Table 1. The total soluble solids (TSS) of the pumpkin samples were detected using a digital Abbe refractometer at 20 °C and the content of the vitamin C was determined using potassium iodate titration [41]. The fruit soluble protein was measured using the protein dye-binding method using bovine serum albumin as a standard [42]. The content of starch was measured using the acid hydrolysis method. Three fruits of each variety were used in these experiments. The results are expressed as the average and its standard deviation (mean ± SD).

### 2.2. Soil Sampling

We used high-throughput sequencing (HTS) methods to study bacterial community structure and diversity in the rhizosphere soils of the different pumpkin varieties grown under the same conditions. The sampling was performed as described previously [43]. For two pumpkin varieties, there were three repetitions. Three plants were randomly harvested as replicates for each pumpkin variety and then we dug out the pumpkin root system, shook off the loose soil particles, and collected the soil attached to the root system as rhizosphere soil [44]. The soil samples were sealed in sterile bags and stored at 4 °C to send to Shanghai Majorbio Bio-pharm Technology Co., Ltd. (Shanghai, China), for PCR amplification and sequencing on an Illumina MiseqPE300 (Illumina, Inc., San Diego, CA, USA) platform.

### 2.3. DNA Extraction and PCR Amplification

The E.Z.N.A.^®^ soil DNA kit (Omega Bio-tek, Norcross, GA, USA) was used in accordance with the manufacturer’s instructions to extract the genomic DNA from the microbial population. The DNA extract was examined on a 1% agarose gel and a NanoDrop 2000 UV-vis spectrophotometer was used to measure the DNA concentration and purity (Thermo Scientific, Wilmington, DE, USA). Using an ABI GeneAmp^®^ 9700 PCR thermocycler, the hypervariable region V3–V4 of the bacterial 16S rRNA gene was amplified using bacterial primers 338F and 806R, while the fungal ITS1 region was amplified using primers ITS1F and ITS2R (ABI, Los Angeles, CA, USA). The specific types and sequences from bacterial and fungal sequencing are shown in Table 2.

The 16S rRNA gene was amplified via PCR in the following manner: initial denaturation at 95 °C for 3 min; then 27 cycles of denaturing at 95 °C for 30 s, annealing at 55 °C for 30 s, and extension at 72 °C for 45 s; followed by a single extension at 72 °C for 10 min, finished at 4 °C. The PCR mixes comprised 2 μL, 2.5 mM dNTPs; 4 μL, 5 × TransStart FastPfu buffer; 10 ng, template DNA; 0.8 μL, forward primer; 0.8 μL, reverse primer; and 0.4 μL, TransStart FastPfu DNA Polymerase, supplemented with ddH_2_O to 20 μL.

The ITS gene was amplified via PCR in the following manner: initial denaturation at 95 °C for 3 min; then 35 cycles of denaturing at 95 °C for 30 s, annealing at 53 °C for 30 s, and extension at 72 °C for 45 s; followed by a single extension at 72 °C for 10 min, finished at 4 °C. The PCR assay was performed using TaKaRa rTaq DNA Polymerase, 20 μL reaction system: 4 μL, 10× Buffer; 2 μL, 2.5 mM dNTPs; 0.8 μL, forward primer; 0.8 μL, reverse primer; 0.4 μL, rTaq Polymerase; 0.2 μL, BSA; and 10 ng, DNA template, created with ddH_2_O to 20 μL.

The triplicate PCRs were carried out. The Axy-Prep DNA Gel Extraction Kit (Axygen Biosciences, Union City, CA, USA) was used to purify the PCR product after it was extracted from 2% agarose gel and a Quantus^TM^ Fluorometer was used to measure the amount (Promega, Madison, WI, USA).

According to the standard methods, the Majorbio Bio-Pharm Technology Co. Ltd. (Shanghai, China). pooled the purified amplicons in an equimolar ratio and sequenced them (2 × 300) on an Illumina MiSeq platform (Illumina, San Diego, CA, USA) (Shanghai, China). The NCBI Sequence Read Archive (SRA) database received the raw reads and assigned the accession number SRP292659 (accessed on 12 November 2020).

### 2.4. Statistics and Analysis

The analyses were carried out using IBM SPSS Statistics 21 and Excel 2019 and a Duncan’s multi-range test was used to compare the mean values. Using mothur (https://mothur.org/wiki/calculators/, accessed on 1 July 2022), investigations of the alpha diversities of soil bacteria and fungi were conducted. Principal coordinate analysis (PCoA) and non-metric multidimensional scaling (NMDS) were performed using R language tools (version 3.3.1, accessed on 4 September 2022). Using OTU tables with a comparable level of 97%, the Venn diagrams were also analyzed using the R language (version 3.3.1, accessed on 29 May 2020) tool. The visual circles of the bacterial and fungi communities in the soil were described using Circos-0.67-7 (http://circos.ca/, accessed on 4 September 2022). The samples were subjected to linear discriminant analysis (LDA) using LEfSe (http://huttenhower.sph.harvard.edu/galaxy/root?tool_id=lefse_upload, accessed on 3 July 2022) to identify species with significant differences from the samples.

## 3. Results

### 3.1. Alpha Analysis

The covering index of all the samples was above 97%, indicating that all the diversity and richness indexes could be used (Table 3). A higher Shannon index indicates higher species richness or evenness; conversely, a lower Simpson index represents higher diversity of the community. Chao1 and ACE indices were also used to represent the richness of the bacterial community.

As seen in Table 3, there was no significant difference in soil bacterial and fungal diversity or richness in the rhizosphere between G1519 and G1511.

### 3.2. Principal Coordinate Analysis (PCoA) and Non-Metric Multi-Dimensional Scaling (NMDS) Analysis

To further evaluate the degree of difference between soil bacterial and fungal communities in the rhizospheres of the two pumpkin varieties (G1519 and G1511), principal coordinate analysis (PCoA) based on the binary Jaccard distance was performed at the OTU level. The results showed that the soil bacterial compositions in the rhizospheres of G1519 were significantly different from those of G1511, with the contribution rate of the first principal coordinate being 31.06% as the main factor affecting the soil bacterial community composition in the pumpkin rhizosphere (Figure 1a). There was a large difference in the soil fungal compositions in the rhizospheres of G1519 and G1511, i.e., the soil fungal compositions of G1519 gathered separately in one quadrant, relatively far from the samples of G1511 (Figure 1c).

Meanwhile, non-metric multi-dimensional scaling (NMDS) was also performed to locate, sort, and classify the sample communities in a dimension-reducing manner, where the distance between sample points indicates the difference or similarity. There was no overlap for both the bacterial and fungal rhizosphere communities of the different pumpkin varieties. The bacterial and fungal compositions in the rhizospheres thus differed significantly between the pumpkin cultivars (Figure 1b,d).

### 3.3. Venn Analysis

As shown in Figure 2a, there were 3230 identical soil bacterial OTUs in the rhizospheres of the low-quality (G1519) and high-quality (G1511) pumpkin varieties. However, the low-quality pumpkin variety (G1519) had only 752 distinct bacterial OTUs, which was quite less than the corresponding number for the high-quality pumpkin cultivar (1060) (Figure 2a).

In addition, the low-quality (G1519) and high-quality (G1511) pumpkin varieties shared 410 rhizosphere soil fungal OTUs. However, the low-quality pumpkin cultivar (G1519) had only 193 OTUs of specific fungi, half the number for the high-quality pumpkin cultivar (398) (Figure 2b).

### 3.4. Soil Microbial Compositions in the Rhizosphere

#### 3.4.1. Soil Bacterial Compositions in the Rhizosphere

As shown in Figure 3a, Proteobacteria, Actinobacteria, Chloroflexi, Firmicutes, Acidobacteria, Gemmatimonadetes, Bacteroidetes, and Patescibacteria were the dominant soil bacterial phyla (i.e., relative abundance > 1%) in the rhizosphere for both G1511 and G1519. Moreover, it is worth noting that Planctomycetes was a unique dominant soil bacterial phylum in G1519. Additionally, the numbers of dominant soil bacteria for the different-quality pumpkin varieties were ten (G1519) and nine (G1511). *Bacillus*, *norank_f__norank_o__norank_c__subgroup_6*, *Streptomyces*, *norank_f__ Gemmatimonadaceae*, *norank_f__JG30-KF-CM45*, *Sphingomonas*, *Pseudomonas*, *Ensifer*, and *Microvirga* were the dominant soil bacterial genera they shared. *Norank_f__A4b* was the only dominant bacterial genus specific to G1519 (Figure 3b).

#### 3.4.2. Soil Fungal Compositions in the Rhizosphere

As seen in Figure 4a, the most abundant soil fungal phylum in the rhizospheres of G1519 and G1511 was Ascomycota, reaching 45.19% and 59.77%, respectively.

Moreover, it is noteworthy that the prevalence of Basidiomycota in the rhizospheric soil of G1519 (39.49%) was much higher than that for G1511 (4.04%). Furthermore, the prevalence of Mortierellomycota in G1511 (5.97%) was lower than that in G1519 (12.09%). Rozellomycota was unique as the dominant fungal phylum in cultivar G1511; on the contrary, Chytridiomycota as the dominant fungal phylum was specific to cultivar G1519 (Figure 4a).

At the genus level, the nine common dominant fungal genera were *Micropsalliota*, *Cephaliophora*, *unclassified_f__Chaetomiaceae*, *Fusarium*, *Penicillium*, *Mortierella*, *unclassified_o__Sordariales*, *unclassified_c__Sordariomycetes*, and *Cladosporium* (Figure 4b).

*Micropsalliota* was much more prevalent in the rhizosphere soil of G1519 (31.01%) than in G1511 (1.59%), and *Fusarium* in G1519 (2.17%) was much less prevalent than in G1511 (6.28%), while the proportion of *Penicillium* in G1519 (1.24%) was much lower than that in G1511 (6.30%). Meanwhile, *Unclassified_p__Rozellomycota* was a unique dominant fungal genus in the rhizosphere of pumpkin cultivar G1511 (27.69%).

### 3.5. LEfSe Multi-Level Species Difference Discriminant Analysis

To identify soil bacterial and fungal differences between the high-quality pumpkin (G1511) and low-quality pumpkin (G1519) cultivars, LEfSe analysis was performed (Figure 5). A total of 14 soil bacterial evolutionary branches showed significant differences (LDA > 3.5), among which *Streptomyces* was significantly enriched in rhizospheres of the G1511 cultivar. By contrast, *norank_f__A4b, Lysobacter,* and *Lysinibacillus* were significantly enriched in rhizospheres of the G1519 cultivar (Figure 5a,b).

In addition, 19 fungal evolutionary branches showed significant differences (LDA > 3.5). At the phylum level, Glomeromycota was significantly enriched in rhizospheres of the high-quality pumpkin cultivar (G1511), and Basidiomycota was significantly enriched in rhizospheres of the low-quality pumpkin cultivar (G1519). At the genus level, *unclassified_o__Microascales* and *Roussoella* were significantly enriched in rhizospheres of the high-quality pumpkin cultivar (G1511); on the contrary, *unclassified_f__Sordariaceae*, *unclassified_f__Cordycipitaceae*, *Spiromastix*, *unclassified_f__ Herpotrichiellaceae*, *Conocybe*, and *Micropsalliota* were significantly enriched in rhizospheres of the low-quality pumpkin cultivar (G1519) (Figure 5c,d).

### 3.6. Network Analysis

#### 3.6.1. Bacterial Network Analysis

Among the soil bacteria, the relative abundance of *norank_f__A4b*, the dominant strain endemic to the low-quality pumpkin cultivar (G1519), was negatively correlated with starch, total soluble solids, and total proteins. *Streptomyces*, with relative abundance levels of 2.90% in G1519 and 4.16% in G1511, was positively correlated with starch, total soluble solids, total proteins, and vitamin C (Figure 6).

#### 3.6.2. Fungal Network Analysis

As shown in Figure 7, *Micropsalliota* was negatively correlated with starch, total soluble solids, total proteins, and vitamin C. Moreover, the relative abundance of *Micropsalliota* was higher in cultivar G1519 (31.01%) than in G1511 (1.59%) (Figure 4b). Additionally, *unclassified_o__Microascales*, *Humicola*, and *Curvularia* of the Ascomycota were each positively correlated with starch, total soluble solids, total proteins, and vitamin C. Among them, the relative abundance of Ascomycota in G1519 (45.19%) was lower than that in G1511 (59.77%) (Figure 4a). Furthermore, Basidiomycota was negatively correlated with starch, total soluble solids, total proteins, and vitamin C (Figure 4b). We found that the relative abundance of Basidiomycota was higher in G1519 (39.49%) than in G1511 (4.04%).

## 4. Discussion

In this study, the soil microbial abundance and diversity in rhizospheres in relation to pumpkin quality were investigated. Firstly, our results revealed that soil microbial diversity (Shannon and Simpson) and richness (Chao1 and ACE) in the rhizospheres of high-quality pumpkins (G1511) and low-quality pumpkins (G1519) were not significantly different. Furthermore, the soil bacterial and fungal compositions in the rhizosphere of high-quality pumpkins (G1511) were significantly different from those for low-quality pumpkins (G1519). This suggests that different genotypes of pumpkin cultivars recruit various bacteria and fungi in their rhizospheres under the same environment and identical management, even though they all belong to *Cucurbita moschata* Duch. One reason can be that distinct plant species or genotypes may have particular phenotypic features, including nutritional compositions, which have a significant role in shaping the compositions of microorganism communities [20,21]. Furthermore, we found that the number of specific soil bacterial OTUs in the rhizosphere of the high-quality pumpkin cultivar (G1511) was 1060—1.42 times higher than that for the low-quality pumpkin cultivar (G1519). Moreover, the number of specific soil fungal OTUs for G1511 was 398—2.06 times higher than that for G1519. These results suggest that the high-quality pumpkin cultivar recruited a greater abundance of soil microbes in its rhizosphere.

In addition, at the phylum level, there were nine and eight dominant soil bacterial phyla in G1519 and G1511, respectively. Among them, Proteobacteria, Actinobacteria, Chloroflexi, Firmicutes, Acidobacteria, Gemmatimonadetes, Bacteroidetes, and Patescibacteria were the common soil bacterial phyla in the rhizospheres of the two pumpkin cultivars. Moreover, the relative percentage of Planctomycetes, the dominant bacterial phylum unique to G1519, was 1.01%. Adam et al. [45] also reported Proteobacteria as the dominant soil bacterial phylum in the pumpkin rhizosphere. Proteobacteria are a group of bacteria commonly found in agroecosystems [46] and they include many of the species responsible for nitrogen fixation [47]. Moreover, Proteobacteria are generally regarded as fast-growing bacteria and are easily adapted to different plant species in plant rhizospheres [48]. Firmicutes contribute to disease-suppressive microbiomes and plants have the property of selecting beneficial microorganisms [49]. The omnipresence of Acidobacteria in natural habitats suggests that they perform several crucial functions. However, the ecology and metabolism of most Acidobacteria are not well understood because they have not been cultured [50]. However, owing to their large abundance in soil, these bacteria might significantly relate to ecosystem health [51]. For example, various phylotypes matched with Acidobacteria Gp1 contribute to the degradation of organics [52,53]. As a result, the ratio of Proteobacteria to Acidobacteria can be used to evaluate the trophic level of soil [54].

Although *Streptomyces* was a dominant soil bacterial genus in the rhizospheres of the G1519 and G1511 pumpkin cultivars, the relative abundance of it in the G1511 rhizosphere was 1.4 times greater than that in the G1519 rhizosphere. *Streptomyces* produces a large number of active secondary metabolites [55,56] that promote plant growth and have biocontrol functions [57] and are also an important source of antagonistic strains of bacteria. Moreover, our study also found that the relative abundance of *Streptomyces* was positively correlated with the total soluble solids, starch, vitamin C, and total protein contents of pumpkin. Our result is consistent with a previous report by Ji et al. [58].

The dominant fungal phyla shared among cultivar G1519 and cultivar G1511 included Ascomycota, Basidiomycota, Mortierellomycota, and Rozellomycota. Among them, Ascomycota and Basidiomycota were the most abundant fungal phyla in the G1519 rhizosphere. By contrast, although Ascomycota was still the most abundant fungal phylum in the G1511 rhizosphere, Basidiomycota was not the second most abundant fungal phylum in this rhizosphere. On the contrary, Rozellomycota was not only unique to the G1511 rhizosphere, but was also the second most abundant fungal phylum in this rhizosphere (27.70%). Ascomycota is the most diverse fungal phylum, comprising the majority of plant pathogens [59], and is the predominant organic matter taxon in agroecosystems [60]. Moreover, Ascomycota constitutes most saprophytic fungi, which grow on and cause the decay of dead organic matter [61]. We also found that the relative abundance of Ascomycota was increased in the rhizosphere soil of the high-quality pumpkin cultivar (G1511), indicating a higher content of organic matter in the G1511 rhizosphere.

At the genus level, *Micropsalliota*, *Cephaliophora*, *unclassified_f__Chaetomiaceae*, *Fusarium*, *Penicillium*, *Mortierella*, *unclassified_o__Sordariales*, *unclassified_c__Sordariomycetes*, and *Cladosporium* were the common fungal genera in the G1519 and G1511 rhizospheres. Among them, Micropsalliota was much more prevalent in the G1519 rhizosphere (31.01%) than in G1511 (1.59%). Furthermore, *Unclassified_p__Rozellomycota* was unique to cultivar G1511 (27.69%). Genotypic factors directly affect the soil microbial community in the rhizosphere [62]. Plants shape their rhizosphere microbiome by stimulating or repressing the abundance of exudates from specific microbial groups [32]. Several kinds of microbial species enriched in the rhizosphere may be recruited by species- and genotype-specific molecular signals secreted from the plant, probably as unique components of the root exudates [21,63]. The plant genotype drives microbial selection in part by depositing specific exudates at the soil–root interface [64,65]. Thus, we speculate that plant genotypes result in different compositions of root exudates, which induce different soil fungal compositions, i.e., different microbial communities in rhizospheres are formed by variances in how microorganisms use the various components of root exudates [66,67]. The various crop species in cropping systems show a variety of growth reactions to the overall soil microbial communities [36]. Therefore, it can be inferred that the formation of pumpkin quality is precisely affected by the soil microbial composition in the rhizosphere of the pumpkin cultivar.

Based on these results, the soil microbial community in the pumpkin rhizosphere could be significantly affected by the genotype. However, only two pumpkin cultivars with high and low nutrient levels in the same soil were selected for our experiment, providing a limited ability to confirm the influence of plant cultivar on the soil microbial community. Further studies are required to analyze more pumpkin cultivars with different nutrient levels.

Additionally, with the development of analysis level, deeper insights into what differences in the rhizosphere soil microbial compositions occur with plant cultivars can be further elucidated; an explanation of the interaction between host plants and their closely related microbiota and the discovery of functional microbes will provide new perspectives illuminating the biological processes in the rhizosphere.

## 5. Conclusions

In comparison with the low-quality pumpkin cultivar (G1519), higher microbial diversity and richness were found in the rhizosphere of the high-quality pumpkin cultivar (G1511). Our results indicate that soil microbes were differentially enriched in the rhizospheres of the low- and high-quality pumpkin cultivars. This also suggests that to develop quality, more abundant soil microbes need to be recruited in the rhizosphere of the high-quality pumpkin cultivar. Rozellomycota and *Unclassified_p__Rozellomycota* may be functional microbes relating to pumpkin quality.

## Figures and Tables

**Figure 1 microorganisms-10-02219-f001:**
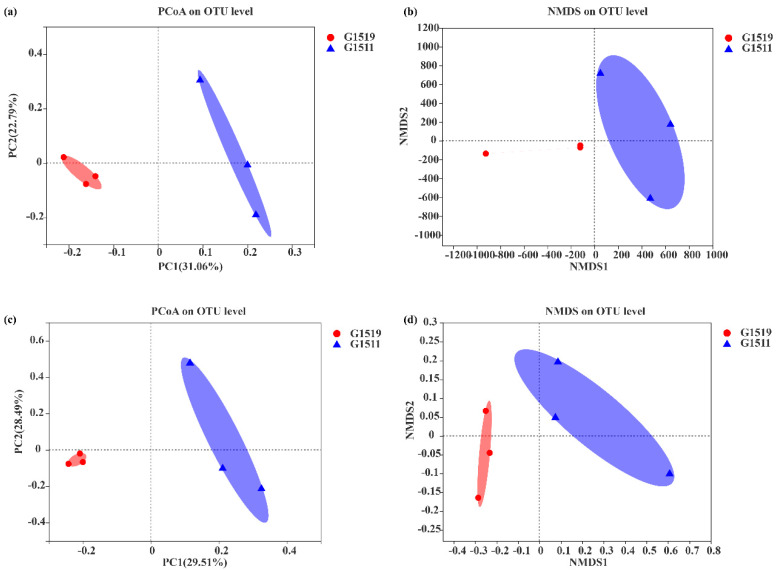
Comparison of soil bacteria and fungi in the rhizospheres of two pumpkin cultivars (G1519 and G1511). (**a**) PCoA analysis of the soil bacterial community at the OTU level. (**b**) NMDS analysis of soil bacterial communities at the OTU level. (**c**) PCoA analysis of soil fungal communities at the OTU level. (**d**) NMDS analysis of soil fungal communities at the OTU level.

**Figure 2 microorganisms-10-02219-f002:**
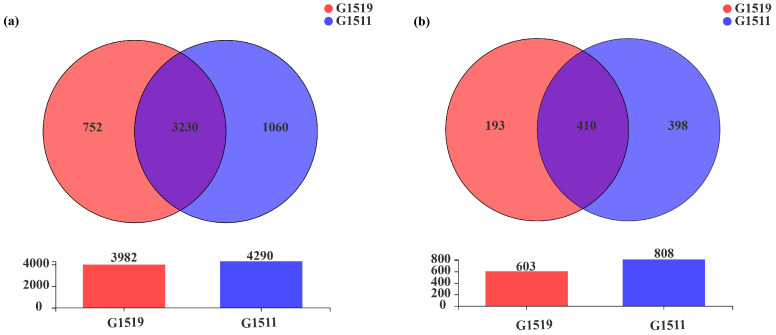
Venn diagram of the soil bacteria (**a**) and fungi (**b**) in the rhizospheres of the two pumpkin cultivars (G1519 and G1511) at the OTU level.

**Figure 3 microorganisms-10-02219-f003:**
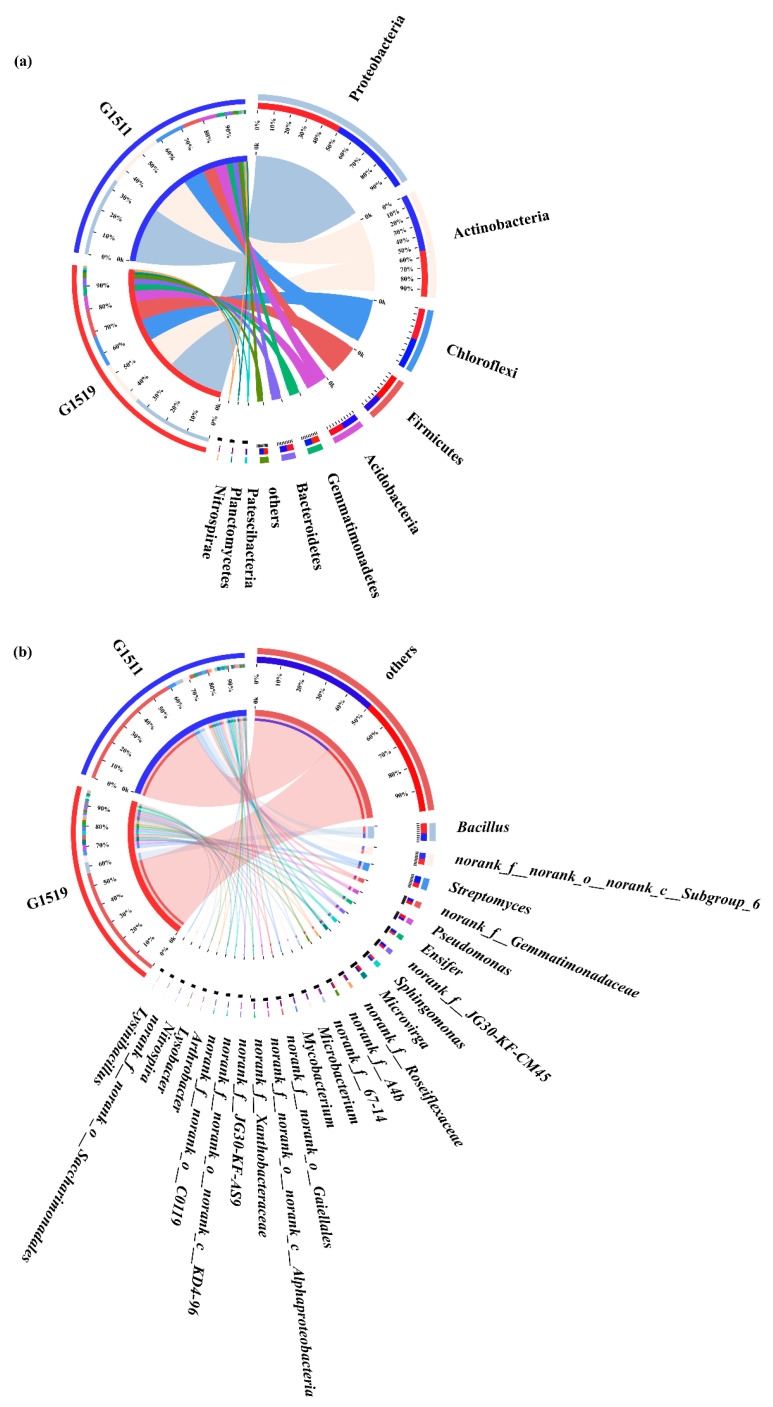
Compositions of soil bacterial communities in the two pumpkin cultivars (G1519 and G1511) at the phylum level (**a**) and genus level (**b**).

**Figure 4 microorganisms-10-02219-f004:**
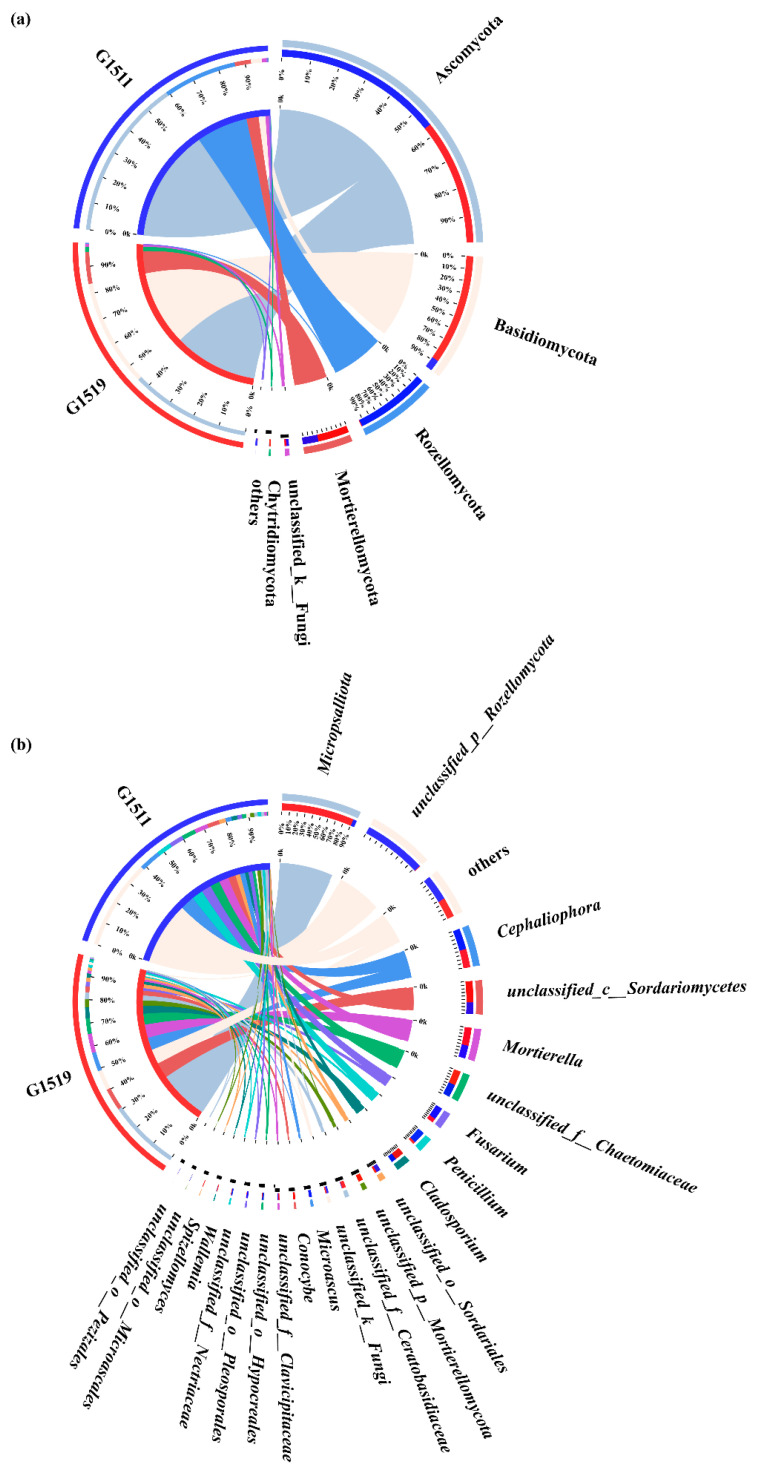
Compositions of soil fungal communities in the two pumpkin cultivars (G1519 and G1511) at the phylum level (**a**) and genus level (**b**).

**Figure 5 microorganisms-10-02219-f005:**
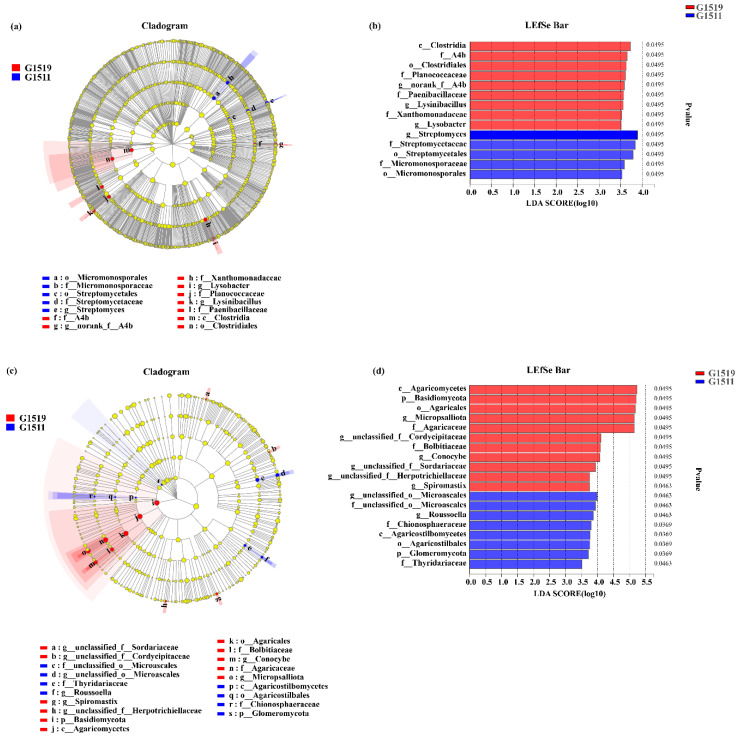
Phylogenetic cladogram of bacterial (**a**) and fungal (**c**) lineages in rhizosphere soil from different pumpkin cultivars (G1519 and G1511). Indicator soil bacteria (**b**) and fungi (**d**) with LDA scores of 3.5 from two pumpkin cultivars.

**Figure 6 microorganisms-10-02219-f006:**
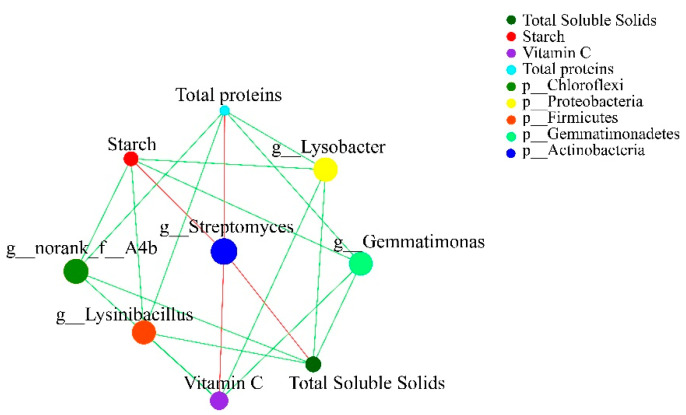
Correlation network analysis of quality characteristics of different pumpkin cultivars and soil bacteria at the genus level (top 50).

**Figure 7 microorganisms-10-02219-f007:**
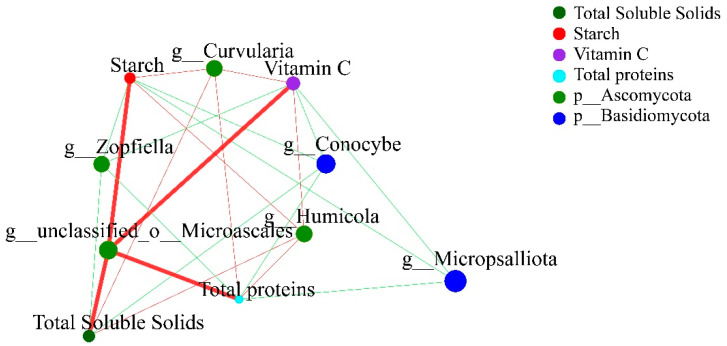
Correlation network analysis of quality characteristics of different pumpkin cultivars and soil fungi at the genus level (top 50).

**Table 1 microorganisms-10-02219-t001:** Quality characteristics of different pumpkin varieties.

Samples	Total Soluble Solids (%)	Starch (g/100 g)	Vitamin C (mg/100 g)	Total Proteins (g/100 g)
G1519	5.8 ± 1.0 b	1.1 ± 0.2 b	7.5 ± 2.5 b	0.9 ± 0.3 b
G1511	7.2 ± 0.7 a	8.0 ± 1.4 a	27.4 ± 7.2 a	1.4 ± 0.3 a

Note that all statistics are presented as the mean ± SD (standard deviation). Significant variations between treatments at *p* < 0.05 are indicated by different letters in the same column.

**Table 2 microorganisms-10-02219-t002:** Sequencing types and primer sequences.

Sequencing Regions	Prime Name	Prime Sequence	Sequencing Platform	Length
16S rRNA gene	338F	5′-ACTCCTACGGGAGGCAGCAG-3′	Miseq	468 bp
	806R	5′-GGACTACHVGGGTWTCTAAT-3′		
ITS gene	ITS1F	5′-CTTGGTCATTTAGAGGAAGTAA-3′	Miseq	300 bp
	ITS2R	5′-GCTGCGTTCTTCATCGATGC-3′		

**Table 3 microorganisms-10-02219-t003:** Comparison of soil bacterial and fungal diversity indices in rhizospheres of G1519 and G1511.

Source	Sample	Shannon	Simpson	Ace	Chao1	Coverage
Soil bacteria	G1519	6.52 ± 0.15 a	0.005 ± 0.001 a	3618.93 ± 215.34 a	3627.73 ± 207.69 a	0.97
	G1511	6.65 ± 0.14 a	0.004 ± 0.001 a	3864.76 ± 235.34 a	3881.39 ± 224.45 a	0.97
Soil fungi	G1519	3.05 ± 0.36 a	0.14 ± 0.07 a	452.46 ± 19.08 a	454.04 ± 23.20 a	0.99
	G1511	3.02 ± 1.75 a	0.25 ± 0.37 a	524.10 ± 162.07 a	513.40 ± 185.30 a	0.99

Note that all statistics are presented as the mean ± SD (standard deviation). Significant variations between treatments at *p* < 0.05 are indicated by different letters in the same column.

## Data Availability

The raw reads were deposited into the NCBI Sequence Read Archive (SRA) database under accession number: SRP292659 (accessed on 12 November 2020).

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
