# Peer review of "Higher-Quality Pumpkin Cultivars Need to Recruit More Abundant Soil Microbes in Rhizospheres"

_microorganisms, 2022, doi:10.3390/microorganisms10112219_

Round 1

Reviewer 1 Report

The presented study focuses on rhizospheric microbial communities of two different pumpkin cultivars. The article submitted for evaluation is well-written and raises an interesting problem, however, the authors have not been avoided errors or understatements.

Introduction:

L53: What are “they”?

LL61-62: This statement requires a citation. In addition, we start a sentence with a capital letter.

Material and Methods:

LL102-105: Superscripts at kg

LL112-113:  Please describe the method for determining vitamin C more precisely.

Paragraph 2.3: Please describe the method of extraction and amplification of fungal DNA

There is no description of the bioinformatics methods used.

No description of the statistical methods used.

Results:

L159: If DNA was extracted from the soil, there could be no concept of endophytic bacterial communities. The term rhizosphere bacteria should be used.

Discussion:

The analysis of bacterial and fungal networks is very interesting. The relationships between groups of rhizosphere microorganisms and the content of various components in pumpkins are fascinating. Therefore, I believe that this part of the research needs a wider discussion.

Conculion:

The conclusions are largely a restatement of the findings in the 'Results' and 'Discussion' sections, this section needs to be rewritten and clearly answered to the stated aims.

Author Response

The presented study focuses on rhizospheric microbial communities of two different pumpkin cultivars. The article submitted for evaluation is well-written and raises an interesting problem, however, the authors have not been avoided errors or understatements.

Introduction:

Point 1: L53: What are “they”?

Response 1: The authors have revised these parts in the manuscript according to the Reviewer’s comments (L53).

Point 2: L61-62: This statement requires a citation. In addition, we start a sentence with a capital letter.

Response 2: Exactly, the authors were sorry to make such a simple mistake. The authors have deleted this part in the manuscript according to the Reviewer’s suggestions (L59-60).

Material and Methods:

Point 3: L102-105: Superscripts at kg

Response 3: The authors have revised these parts in the manuscript according to the Reviewer’s comments (L101-103).

Point 4: L112-113:  Please describe the method for determining vitamin C more precisely.

Response 4: The authors have revised these parts in the manuscript according to the Reviewer’s comments (L109-110).

Point 5: Paragraph 2.3: Please describe the method of extraction and amplification of fungal DNA

Response 5: The authors have revised these parts in the manuscript according to the Reviewer’s comments (L134-135).

Point 6: There is no description of the bioinformatics methods used.

No description of the statistical methods used.

Response 6: Analyses were carried out using IBM SPSS Statistics 21 and Excel 2019, and Dun-can's multi-range test was used to compare mean values. Mothur (version v.1.30.2, available at https://Mothur.org/wiki/calculators/) was used for Principal Co-Ordinates Analysis (PCoA) and Non-metric Multi-dimensional Scaling (NMDS). The R language (version 3.3.1) tool was used to identify and plot an OTU table with a level of 97% sim-ilarity for microbial community composition and Venn diagram analysis. LEfSe samples were submitted to linear discriminant analysis (LDA) to find clusters that had signifi-cantly different effects on sample delineation (http://huttenhower.sph.harvard.edu/glaxy/root? Tool id = LEfSe pload). NetworkX (https://networkx.org/) was used as a toolset for co-existing network analysis. The authors have revised these parts in the manuscript according to the Reviewer’s comments (L152-162).

Results:

Point 7: L159: If DNA was extracted from the soil, there could be no concept of endophytic bacterial communities. The term rhizosphere bacteria should be used.

Response 7: Exactly, authors have deleted this part in the manuscript according to the Reviewer’s suggestions (L166).

Discussion:

Point 8: The analysis of bacterial and fungal networks is very interesting. The relationships between groups of rhizosphere microorganisms and the content of various components in pumpkins are fascinating. Therefore, I believe that this part of the research needs a wider discussion.

Response 8: The authors have revised these parts in the manuscript according to the Reviewer’s comments (L371-374).

Conclusion:

Point 9: The conclusions are largely a restatement of the findings in the 'Results' and 'Discussion' sections, this section needs to be rewritten and clearly answered to the stated aims.

Response 9: The authors have revised these parts in the manuscript according to the Reviewer’s comments (L388-396).

Special thanks to you for your good comments.

Reviewer 2 Report

The work is data rich -- lots of information about genera etc  

However it is in isolation-  could be improved by examination of root exudate composition  

Also i am not sure how many replicates for each treatment  -may have overlooked in my reading

what would change if samples were at a different time point

One set of data that i feel is essential is the same analysis of bulk soil 

Author Response

Point 1: The work is data rich -- lots of information about genera etc

However, it is in isolation- could be improved by examination of root exudate composition

Response 1: In fact, root exudates are the direction of our future investigation. We found that the composition and content of root exudates were also affected by soil microorganisms.

Point 2: Also, I am not sure how many replicates for each treatment -may have overlooked in my reading

Response 2: Each treatment with three repetitions, and this part is discussed in L121-121.

Point 3: what would change if samples were at a different time point

Response 3:  The results in this manuscript were based on the samples which collected randomly at a time point. Therefore, the authors inferred that it would be the same trends at different time points. But it might be need to further confirmed.

Point 4: One set of data that I feel is essential is the same analysis of bulk soil

Response 4: Exactly, the bulk soil also might be affected by different quality pumpkin cultivars. However, in comparison with the rhizospheric soils of different pumpkin cultivars, higher sensitivity of responses will be observed in rhizospheric soils. Therefore, the authors mainly focused on the soils in rhizospheres only in this manuscript.

Comments made by the reviewers in peer-review-23302422 have been individually revised by the authors and marked in red in the manuscript.

Special thanks for your good comments, Thank you!

Reviewer 3 Report

General comments

the research article entitled “Higher quality pumpkin cultivars need to recruit more abundant soil 2 microbes in rhizospheres” investigates microbial community structure of two pumpkin cultivar (low and high quality) grown under similar management regime. Although interesting, the study lacks novelty as it is already known that different cultivar of a same plant will recruit different rhizosphere microbes. I do not agree with the research conclusion stating that quality traits of pumpkin could be attributed to some specific microorganisms and there are two reasons for that. 1. Microbial functionality was not assessed (richness do not necessarily imply functions). 2.  Multiple soil types (with respect to the microbial profile including a sterile soil) should have been assayed which may enable to establish a trend regarding the correlation between the richness of specific microbial species and quality traits. Based on the provided result. Improved quality could be attributed to the genetic properties of the cultivar, microbial community or both. A strongly suggest reviewing the research hypothesis and conclusion.

Specific remarks    

English language should be reviewed. There are several grammars and spelling mistakes

Replace Unclassified_p__Rozellomycota  with Unclassified Rozellomycota (make change in the whole text for other species)

Line 110: What do you mean by conventionally managed? Provide fertilization and irrigation regime

Table 1: title do not correspond to data

Figure 3: figure is not clear. increase resolution or replace it with classic histogram

For the correlation analysis provide value of pearson correlation (the most significant)

The discussion section should be completely reviewed. Hypothesis and results should be further developed and compared to previous relevant research report  

Author Response

General comments

Point 1: the research article entitled “Higher quality pumpkin cultivars need to recruit more abundant soil 2 microbes in rhizospheres” investigates microbial community structure of two pumpkin cultivar (low and high quality) grown under similar management regime. Although interesting, the study lacks novelty as it is already known that different cultivar of a same plant will recruit different rhizosphere microbes. I do not agree with the research conclusion stating that quality traits of pumpkin could be attributed to some specific microorganisms and there are two reasons for that. 1. Microbial functionality was not assessed (richness do not necessarily imply functions). 2.  Multiple soil types (with respect to the microbial profile including a sterile soil) should have been assayed which may enable to establish a trend regarding the correlation between the richness of specific microbial species and quality traits. Based on the provided result. Improved quality could be attributed to the genetic properties of the cultivar, microbial community or both. A strongly suggest reviewing the research hypothesis and conclusion.

Response 1: The authors have revised these parts in the manuscript according to the Reviewer’s comments (L389-396).

Specific remarks

Point 2: English language should be reviewed. There are several grammars and spelling mistakes

Response 2: We agree with this suggestion and have modified terminology throughout the text as appropriate.

Point 3: Replace Unclassified_p__Rozellomycota with Unclassified Rozellomycota (make change in the whole text for other species)

Response 3: We respectfully disagree because the "p" in Unclassified_p__Rozellomycota refers to phylum classification.

Point 4: Line 110: What do you mean by conventionally managed? Provide fertilization and irrigation regime

Response 4: The authors have revised in “identically managed” in manuscript according to the Reviewer’s suggestion (L106-107).

Point 5: Table 1: title do not correspond to data

Response 5: Exactly, the authors were sorry to make such a simple mistake. The authors have deleted this part in the manuscript according to the Reviewer’s suggestions (L114).

Point 6: Figure 3: figure is not clear. increase resolution or replace it with classic histogram

Response 6: The authors have revised these parts in the manuscript according to the Reviewer’s comments (Figure 3).

Point 7: For the correlation analysis provide value of pearson correlation (the most significant)

Response 7: The authors have revised these parts in the manuscript according to the Reviewer’s comments (Figure 5).

Point 8: The discussion section should be completely reviewed. Hypothesis and results should be further developed and compared to previous relevant research report

Response 8: The authors have revised these parts in the manuscript according to the Reviewer’s comments (L371-375).

Special thanks to you for your good comments.

Round 2

Reviewer 1 Report

Dear Authors,

The manuscript submitted for re-evaluation still requires improvements. My concerns are listed below:

LL137-150: The amplification process of the16S rRNA gene has been described in detail, while a description of the amplification conditions of the ITS gene is still missing.

The bioinformatics part is still perfunctory and requires a full description of the analyses performed. Otherwise, we are unable to ascertain the correctness of the results obtained.

LL371-374: I had expected a broader discussion on this interesting topic.  The issue of relationships between groups of rhizosphere microorganisms and the content of various components in pumpkins certainly needs a more detailed explanation.

Reviewer 2 Report

Please please please take editing seriously. The quality of this work is degraded by poor writing and organization. 

this work is very factual  and i am not convinced that your data set tells the story

I would like to see how different the three pumpkins of one type were   when you average everything you can get wrong conclusions

if the pumpkins were close in the field  maybe similar     

so guess i am asking for more facts 

Reviewer 3 Report

The manuscript has been significantly improved and most of the comments have been addressed. I recommend for publication after English language checking (minor)
